# Anti-Cyclic Citrullinated Peptide Antibody Index in the Cerebrospinal Fluid for the Diagnosis and Monitoring of Rheumatoid Meningitis

**DOI:** 10.3390/biomedicines10102401

**Published:** 2022-09-26

**Authors:** Luigi Caputi, Giorgio B. Boncoraglio, Gaetano Bernardi, Emilio Ciusani, Marcello Dantes, Federica de Liso, Alessandra Erbetta, Gianluca Marucci, Caterina Matinato, Elena Corsini

**Affiliations:** 1Neurology Unit, Department of Cardio-Cerebrovascular Diseases, Maggiore Hospital ASST-Crema, 26013 Crema, Italy; 2Department of Cerebrovascular Disease, Fondazione IRCCS Istituto Neurologico Carlo Besta, 20133 Milan, Italy; 3Department of Diagnostic and Technology, Fondazione IRCCS Istituto Neurologico Carlo Besta, 20133 Milan, Italy; 4Department of Reumatology, IRCCS Humanitas Research Hospital, 20089 Rozzano, Italy; 5Laboratory of Clinical Chemistry and Microbiology, Fondazione IRCCS Ca’ Granda Ospedale Maggiore Policlinico, 20122 Milan, Italy; 6Neuroradiology Unit, Department of Technology and Diagnosis, Fondazione IRCCS Istituto Neurologico Carlo Besta, 20133 Milan, Italy; 7Neuropathology Unit, Fondazione IRCCS Istituto Neurologico Carlo Besta, 20133 Milan, Italy

**Keywords:** neurological autoimmune disorders, rheumatoid meningitis, anti-cyclic citrullinated peptide antibody

## Abstract

Rheumatoid meningitis (RM) is a rare but often aggressive neurological complication of rheumatoid arthritis. The diagnosis of RM, besides the clinical, radiological, and laboratory criteria, usually requires a cerebral biopsy. Based on the two cases presented in this paper, we propose a new laboratory marker. Cerebrospinal fluid and serum anti-cyclic citrullinated peptide (CCP) IgG were measured, and the intrathecal synthesis of anti-CCP antibodies (anti-CCP antibody index) was calculated using the hyperbolic function. The anti-CCP antibody index was positive in both cases at first diagnosis and progressively decreased after treatments. Together with clinical and radiological criteria, the calculation of the anti-CCP intrathecal synthesis, more than the simple measurement of serum or cerebrospinal fluid anti-CCP antibody titers, may represent a useful tool for RM diagnosis and, possibly, for treatment response.

## 1. Introduction

Rheumatoid arthritis (RA) is a multiorgan inflammatory disorder with a prevalence of approximately 0.5–1% in the general population [1]. Although the involvement of the central nervous system is uncommon, rheumatoid meningitis (RM), involving both pachymeninges and leptomeninges, is a potentially aggressive manifestation of RA [2,3,4]. In a recent review, less than 100 cases of RM are described in the literature [3].

RM may mimic several different neurological diseases, infections, and malignancies; therefore, it is often clinically non-specific and difficult to recognize, particularly without a previous RA diagnosis [4,5,6,7,8]. Contrast-enhanced brain MRI, together with the evaluation of serum rheumatoid factor (RF), anti-cyclic citrullinated peptide (anti-CCP) antibody titers, and the cerebrospinal fluid (CSF) analysis, may help in diagnosis [9]. However, for a definite RM diagnosis, a meningeal biopsy showing rheumatic noduli is recommended [2,10,11,12,13,14], and it is performed in nearly three out of four cases [4]. Due to the invasive nature of the meningeal biopsy, some reports have recently focused on the evaluation of RF and/or anti-CCP antibody titers in the CSF [7,9,10,14,15,16,17,18,19,20,21,22,23]. Interestingly, some of these studies showed a reduction of CSF RF and/or anti-CCP antibody titers after specific treatments, indicating these tests are potential biomarkers useful for diagnosis and the evaluation of treatment response [9,17,19,21,23].

Since immunoglobulin (Ig) concentration in CSF is a function of serum Ig concentration, blood–brain barrier integrity, and intrathecal synthesis [24], in addition to the simple titer of anti-CCP antibody in serum and CSF, we calculated the amount of anti-CCP antibody intrathecal synthesis by the Reiber formulas. Using this approach, we here describe two patients with RM displaying significant anti-CCP intrathecal synthesis that normalized after treatment. Hence, these results would state that anti-CCP antibodies are intrathecally synthesized during the active phases of RM, and their quantitation might be a useful tool for diagnosis and follow-up. Furthermore, we reviewed 26 cases of RM for which a specific evaluation in terms of RF, anti-CCP antibodies, and/or intrathecal Ig synthesis in CSF was assessed.

## 2. Materials and Methods

Patients underwent lumbar puncture at time of admission (T0) and follow-up (T1, T2, and T3 after 9, 18, and 28 months for Case 1 and after 5, 13, and 19 months for Case 2). The following CSF parameters were examined: glucose, total protein, and IgG were measured using an automated analyzer (Architect; Abbott; Illinois, USA); IgA and IgM were assessed by Optilite (Binding Site; Birmingham, UK); cells were counted using a Fuchs–Rosenthal counting chamber; and detection of oligoclonal IgG bands was performed by isoelectrofocusing, as previously described [25].

Intrathecal synthesis was calculated for IgG, IgM, IgA, and anti-CCP antibodies according to Reiber formulas [24]. Kappa and free light chain indexes were calculated, as described elsewhere [26].

### 2.1. Specific Antibody Index

Anti-CCP antibodies were determined using the third-generation assay Quanta Flash™ anti-CCP3 IgG kit (BIOFLASH Rapid Response Chemiluminscence Analyzer; INOVA Diagnostics; San Diego, CA, USA).

To detect the brain-derived fraction of anti-CCP antibodies, the antibody index (AI) was calculated with the following formulas [24]:Anti CCP AI = anti-CCP Quotient/IgG Quotient (normal range: 0.5–1.5)
where
anti-CCP Quotient = anti-CCPCSF/anti-CCPSERUM
IgG Quotient = IgG_CSF_/IgG_SERUM_

If IgG Quotient > QLim, anti-CCP AI was calculated as follows:Anti CCP AI = anti-CCP Quotient/QLim
where
QLim = 0.93 ∗ (QAlb2 + 6 ∗ 10^−6^) 0.5 − 1.7 ∗ 10^−3^

Instead of the simple detection of serum and/or CSF anti-CCP antibody titers, this approach allows the distinction, in the CSF, between the amount of anti-CCP antibody passively diffused in the brain out of the serum through the blood–CSF barrier from the fraction produced within the CNS by resident plasma cells (if any). Theoretically, in the absence of intrathecal synthesis, the AI values calculated by the described formulas should be 1; however, due to several factors (such as inter-individual variability and specific test coefficient variation), AI values lower than 1.5 are considered physiological [24]. These well-known formulas are currently used to evaluate intrathecal synthesis in viral and autoimmune diseases and to detect polispecific immune response to the three neurotropic viruses (measles (M), rubella (R), and varicella zoster (Z)) in multiple sclerosis patients (the so-called “MRZ reaction”) [24,27,28].

### 2.2. Case Descriptions

#### 2.2.1. Case 1

A 74-year-old woman was admitted to our institute in January 2017. She reported a regular lifestyle, arterial hypertension, and, in both eyes, dystrophy of retinal pigment epithelium, cataracts, and dry-eye syndrome. In October 2016, she had flu-like syndrome with a severe headache, followed by left-side weakness and numbness. Brain CT was normal, and carotid Doppler revealed a 50% right carotid stenosis; she was discharged with diagnosis of minor ischemic stroke. In November 2016, she experienced clonic movements involving the lower left limb and started levetiracetam 500 mg twice/day and pregabalin 50 mg at night. Brain contrast MRI (Figure 1A–E) revealed subacute subarachnoid blood suffusion in right frontal convexity. Thrombophilic screening, HIV test, serum Angiotensin Converting Enzyme, and QuantiFERON-TB were normal, while autoimmunity screening showed normal CRP, increased RF (66.9 UI/mL), increased anti-CCP antibodies (340 UA), and positive ANA (1:160). The patient had full control of focal seizures but still reported headaches, mild fever, and left leg weakness. She also reported pain, redness, and swelling in the right wrist. In January 2017, she was again admitted to our institute: all previous findings were confirmed. CSF analysis revealed essentially normal proteins, increased WBCs (58/μL; 90% lymphocytes), and presence of oligoclonal bands. Full-body FDG-PET and CT were unremarkable, while wrist ultrasound examination showed bilateral micro-erosions of the opposite bone heads of the ulno-carpic joint with a very thin spillage. These findings were considered suggestive of rheumatoid arthritis and meningitis. Given the very mild neurological symptoms, we excluded the brain biopsy and treated the patient with high doses of IV methylprednisolone followed by slow tapering to prednisone 5 mg and methotrexate 15 mg/week.

After discharge, the headache improved. Two months later, the patient had pain and swelling in the right knee. Arthrocentesis showed 0.305 WBC (90% mononuclear cells) and 0.003 RBC. Methotrexate was increased to 20 mg/week.

In October 2017, the patient did not complain of any neurological symptoms, and we observed the remission of the CSF abnormalities; MRI pattern appeared stationary. However, due to the worsening of the RA, methotrexate was substituted with salazopyrin 500 mg twice/day, continuing prednisone 5 mg/day.

In February 2018, salazopyrin was substituted with hydroxychloroquine 200 mg/day.

In July 2018, from both a neurological and neuroradiological point of view, no significant changes were observed. CSF inflammatory markers worsened slightly for the presence of oligoclonal bands, while CSF proteins and cells were normal.

At the last check-up in May 2019, brain neuroimaging appeared stable, while the clinical and CSF findings were normal (see Table 1), continuing prednisone 5 mg and hydroxychloroquine 200 mg/day.

#### 2.2.2. Case 2

A 46-year-old woman was admitted to our institute in October 2017. At the age of 33, after childbirth, she had a right fronto-parietal brain hemorrhage due to cerebral venous thrombosis, followed by deep venous thrombosis and pulmonary thromboembolism. She started anticoagulants, and since no specific thrombophilia was observed, she was going to stop the treatment when she had relapses of deep venous thrombosis; therefore, she went on with long-lasting anticoagulants. Three months after cerebral venous thrombosis, the patient had her first seizure with focal motor onset at the left hand and started carbamazepine. Brain MRI showed the lesions related to the previous hemorrhage that occurred three months earlier. During the following 10 years, she had approximately two focal seizures per year, and the treatment was not modified.

From February to September 2017, she had several focal seizures with motor onset at the left lower and upper limbs, sometimes with possible impaired awareness, which encouraged the caregivers to modify the antiepileptic therapy by increasing carbamazepine and introducing levetiracetam. The patient also underwent a contrast-enhanced brain MRI (Figure 1F–J) that showed a leptomeningeal enhancement in the supratentorial regions, mostly of the right hemisphere, at the convexity and within the interhemispheric fissure. A superficial siderosis close to the right fronto-parietal brain hemorrhage was observed. A few weeks later, the patient was admitted to our institute. The neurological examination was normal; normal were also routine blood tests; screening for thrombophilia; serum ACE and onconeural antibodies; QuantiFERON-TB; neoplastic markers; IgG 4 amount; antiviral antibody responses to CMV, EBV, HSV-1/2, and HIV; and antibody responses to Borrelia Burgdorferi and Treponema. Screening for autoimmune diseases showed an increase in serum CRP (5.1 mg/dL), erythrocyte sedimentation rate (80 mm), RF (50.8 UI/mL), and anti-CCP antibodies (485 UA). CSF revealed an increase in WBC (87 cells/μL; 95% lymphocytes) with normal protein and glucose amounts. Oligoclonal bands were present (pattern type 3), and no intrathecal IgG synthesis for CMV, EBV, HSV-1/2, Treponema, and Borrelia was detected. Interestingly, anti-CCP index (43.6) was extremely high, indicating intrathecal synthesis of anti-CCP antibodies. A contrast-enhanced brain MRI was repeated in October 2017, and an increase in the leptomeningeal enhancement in the left supratentorial regions was observed. Malignancy was ruled out by whole-body FDG-PET and CT scan. A moderate increase in glucose uptake, in terms of inflammatory pattern, was observed mostly within the lungs and lymphnodes, and no surgical indication for a loco-regional biopsy was given. We postulated neurosarcoidosis vs. rheumatoid leptomeningitis, even without any sign or symptom related to RA. A meningeal biopsy was proposed, but the patient refused it, accepting an IV methylprednisolone 1 g/day for five days, followed by prednisone 1 mg/kg per os with slow tapering. The patient, seizure-free and under prednisone 5 mg/day, underwent a contrast-enhanced brain MRI in March 2018. The leptomeningeal enhancement was reduced in the right hemisphere, whereas it was more evident in the left frontal lobe. CSF was repeated and results essentially confirmed the previous increase in WBC counting (82 cells/μL; 95% lymphocytes), normal protein and glucose amount, the presence of oligoclonal bands (pattern type 2), and anti-CCP antibody index (AI = 76.1). Serum ACE was normal, and anti-CCP antibodies were still high (113.5 UA). In April 2018, the patient had a tonic-clonic seizure with possible focal motor onset, after which the daily dosage of levetiracetam and oral prednisone were increased.

In April 2018, the patient agreed to undergo cerebral biopsy. Histological examination showed meningeal tissue exhibiting thickening, fibrosis, and focal meningothelial hyperplasia with florid inflammatory infiltration, mostly composed by lymphocytes, plasma cells, and scattered epithelioid cells. Necrotizing granulomas delimited by histiocytes and epithelioid cells were present (Figure 2). The search for Mycobacterium Tuberculosis DNA on the bioptic sample was negative. These neuropathological findings were consistent with the diagnosis of RM.

The patient started IV cyclophosphamide (0.7 g/m^2^ per month for 6 months) up to October 2018, in addition to oral prednisone (10 mg/day). She underwent her third CSF evaluation in November 2018. WBC were considerably reduced (6 cells/μL, 85% lymphocytes); protein and glucose were normal; no evidence of oligoclonal bands; and the anti-CCP antibody index was in the normal range. Serum anti-CCP antibodies were 104.4 UA and RF was 51.8 IU/mL. Contrast-enhanced brain MRI showed reduced enhancement in the left precentral region and increased enhancement in the left fronto-parietal region. A few days later, the patient had a prolonged episode of drowsiness and apathy which was related to possible seizure and post-ictal state. Lacosamide was therefore added with slow tapering of carbamazepine. Taking into account the improvement of CSF data and the MRI data (which never showed a clear reduction of the leptomeningeal enhancement despite the treatments and the persistence of seizures, even if at lower frequency), we decided to start therapy with MTX 15 mg/week plus oral folic acid 10 mg/week and prednisone 10 mg/day. Brain MRI performed in May 2019 showed the persistence of leptomeningeal enhancement in both hemispheres, mostly on the left side. CSF showed WBC 1.6 /μL, with normal protein and glucose and no oligoclonal bands. Serum RF and anti-CCP antibodies were both increased to 17.9 IU/mL and 65 UA, respectively (Table 1). MTX dosage was increased up to 25 mg/week with prednisone 7.5 mg/day. In June 2019, MTX was decreased to 20 mg/week because of the occurrence of nausea. The last brain MRI was performed in January 2020 and appeared stable compared to the one performed eight months earlier. The patient has been seizure-free since November 2018, with oral prednisone 5 mg/day.

#### 2.2.3. Control

A 71-year-old woman affected by neurodegenerative disorder and RA without RM was referred to our institute in 2019. Although serum anti-CCP antibodies value was elevated (405 UA), the anti-CCP index was notably normal (AI = 0.5).

## 3. Discussion

Within the present study, we describe two cases of RM, including neurological evaluations, brain MRI, CSF examination, and, for Case 2, cerebral biopsy, followed for 28 and 19 months, respectively. Both cases developed RM before RA (at the last follow-up, Case 2 did not develop RA at all), and the elevated serum anti-CCP antibodies were very helpful in supporting the diagnosis of RM.

We found 26 cases of RM published in the English literature in the past 20 years, for which a specific evaluation in terms of RF, anti-CCP antibodies in CSF, and/or intrathecal Ig synthesis was assessed [7,9,10,11,14,15,16,17,18,19,20,21,22,23,29,30,31,32,33,34]. Most of these studies showed the presence of non-specific Ig synthesis in CSF, CSF-restricted oligoclonal bands, and an increase in cell and/or protein amount [9,10,19,21,22,23,29]. Some of these studies evaluated CSF anti-CCP antibodies at the onset, and four of them followed up their titers after the treatment [9,19,21,23]. Interestingly, some authors [9,21,23] showed a concomitant reduction in serum and CSF anti-CCP Ab titers at the follow-up with steroid plus Rituximab, steroid plus methotrexate, and steroid plus cyclophosphamide and salazosulfapyridine, respectively. Moreover, Nissen et al. reported both the disappearance of CSF-restricted oligoclonal bands and the reduction of non-specific Ig synthesis at the follow-up [9]. Only one study calculated the anti-CCP AI using the Reiber formula, and our results were comparable [23].

In cohort studies that investigated second-generation anti-CCP antibodies (in this study, we used third-generation assays) in the serum of patients with early RA, sensitivity, and specificity were 57% (95% CI, 51% to 63%) and 96% (CI, 93% to 97%), respectively [35]. The presence of anti-CCP antibodies may precede the development of RA and identify patients at increased risk of a severe disease course [1,35,36].

In RA patients, the serum anti-CCP IgG titer does not usually correlate with disease activity and/or treatment response [36], while in the two RM cases reported here, we observed the progressive reduction of serum anti-CCP IgG in response to specific treatments. However, compared to serum anti-CCP IgG titers, the evaluation of anti-CCP AI was more strictly related to the clinical course of the disease. In fact, anti-CCP AI returned within the normal range after treatment, suggesting that this might represent a useful tool for the diagnosis and management of patients with RM. The calculation of the anti-CCP antibody index might increase the sensitivity and specificity of laboratory diagnostics for RM and, therefore, reduce the need to perform a more invasive cerebral biopsy. Indeed (since the amount of anti-CCP antibodies in CSF may be due to a passive diffusion from serum, blood–CSF barrier dysfunction, and eventual intrathecal synthesis), by the use of this index it is possible to identify if a fraction of the CSF anti-CCP antibodies is produced within the central nervous system, indicating local inflammation. This approach has been used for a long time in the study of viral infections or to detect polispecific immune responses in multiple sclerosis [24,27,28]. Notably, the normal anti-CCP antibody index that we found in a woman affected by RA with elevated serum anti-CCP antibodies but without RM, further suggests that anti-CCP intrathecal synthesis is pathognomonic of RM and would be detectable only in patients with the active disease. Our findings are in agreement with recently reported data showing the intrathecal synthesis of anti-CCP antibodies in RM (anti-CCP index 2.5) which decreased at the 90-day follow-up (<1.8) after treatments [23].

In our patients, the anti-CCP AI was not directly related to serum anti-CCP antibody values. Indeed, while serum anti-CCP antibody titers remained elevated in patient 1 until T3 and in patient 2 until T2, the anti-CCP antibody index turned negative after treatments (T2) of both patients (see Table 1). Together with clinical data, this suggests that systemic and central nervous system diseases follow a different course. Anti-CCP AI is a better marker of RM than serum or CSF anti-CCP titer, thus addressing central nervous system involvement more specifically and eventually allowing more aggressive treatments.

We did not systematically evaluate RF in CSF in our patients; however, serum RF was reduced during the follow-up, as expected and previously reported [9,17,19,21,23]. During the follow-up, an intrathecal synthesis of IgM was observed in both patients, as well as elevated Kappa and Lambda indexes. These findings seem to suggest a subclinical persisting inflammatory process in the central nervous system. As a matter of fact, in line with what was recently described, our patient’s brain MRI also improved at the follow-up, even if it was not completely resolved [20,32]. This would point out the need for chiefly taking into account the CSF data and the clinical evaluation to better characterize the prognosis.

Although with the intrinsic limitation of the small number of patients, our data highlights and corroborates the assumption that the evaluation of intrathecal synthesis of anti-CCP antibodies, more than the anti-CCP antibodies themselves, might be used for diagnosing and monitoring RM. Meningeal biopsy is still the gold standard for RM diagnosis, but detection of CCP intrathecal synthesis could confirm CNS involvement when this invasive medical procedure can not be performed.

## Figures and Tables

**Figure 1 biomedicines-10-02401-f001:**
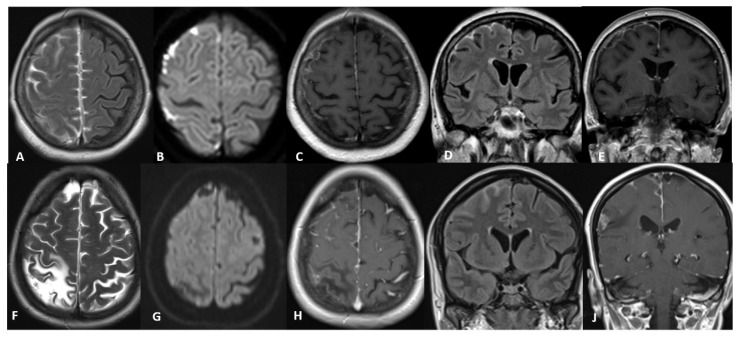
Case 1, (**A**–**E**): axial (**A**) and coronal (**D**) Fluid Attenuated Inversion Recovery (FLAIR) images show hyperintensities within the sulci without edema and mass effect in the fronto-parietal region and in the interhemispheric fissure. Axial (**C**) and Coronal (**E**) T1-weighted images with gadolinium show leptomeningeal enhancement. Note in diffusion-weighted image (**B**) abnormal diffusivity within the sulci. Case 2, (**F**–**J**): axial (**F**) T2-weighted image shows a fronto-parietal poroencephalic lesion on the right side due to venous thrombosis that occurred 10 years earlier. Coronal (**I**) FLAIR image demonstrates leptomeningeal hyperintensities, and axial (**H**) and coronal (**J**) T1-weighted images with gadolinium show leptomeningeal enhancement on both convexities, mostly on the right side. Patient 2 axial diffusion-weighted image (**G**) does not show abnormalities after therapy.

**Figure 2 biomedicines-10-02401-f002:**
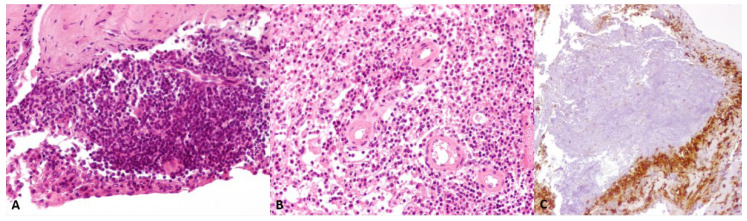
(**A**): meningeal tissue showed thickening, fibrosis, focal meningothelial hyperplasia, and inflammatory infiltration, with scattered epithelioid cells (H&E, 200× magnification). (**B**): the florid inflammation consisted predominantly of lymphocytes and plasma cells (H&E, 200× magnification). (**C**): necrotizing granulomas containing debris of neutrophils were delimited by numerous CD68 immunostained histiocytes and epithelioid cells (200× magnification).

**Table 1 biomedicines-10-02401-t001:** Cerebrospinal fluid parameters in Case 1 and Case 2.

	CSF Protein mg/dL (10–45)°	Cells Cell/μL (<4/μL)°	Oligoclonal Bands (Negative)°	Serum Anti-CCP UA (0–20)°	Anti-CCP AI (0.5–1.5)°
Case 1 (T0)	48.7	58	Positive	536.0	43.7
T1 (9 m.)	33.8	1	Negative	269.9	4.7
T2 (18 m.)	37.7	4	Positive	208.0	0.63
T3 (28 m.)	36.2	1.6	Negative	314.3	<0.5
Case 2 (T0)	33	87	Positive	485.3	43.6
T1 (5 m.)	26.3	82	Positive	113.5	76.1
T2 (13 m.)	20	6	Negative	104.4	<0.5
T3 (19 m.)	19	1.6	Negative	65	<0.5

° normal range

## Data Availability

Not applicable.

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
