# Peer review of "Anti-Cyclic Citrullinated Peptide Antibody Index in the Cerebrospinal Fluid for the Diagnosis and Monitoring of Rheumatoid Meningitis"

_biomedicines, 2022, doi:10.3390/biomedicines10102401_

Round 1
Reviewer 1 Report
Caputi et al's manuscript describes two rare cases of rheumatoid meningitis (RM) is a rare but often aggressive neurological complication of rheumatoid arthritis. Based on the two cases presented in this paper, they propose a new laboratory marker, anti-cyclic citrullinated peptide (CCP) IgG, which was measured in the serum and CSF. They calculated an anti-CCP index. Overall, the two cases are well prepared and the manuscript and easy to read. I have the following comments. After all, the authors write that the RM is a rare event. Are there reliable figures on how often this occurs? You yourself describe 26 cases from the literature in 20 years. That is very few. As controls, wouldn't RA patients without proven RM be better? This also raises the question of how high the anti-CCP AK value is in the CSF of these patients. This would be important evidence for the significance of the anti-CCP titer. The same is true for neurological inflammatory diseases like multiple sclerosis. Here a comparison would be very interesting. It would have to be worked out how high a healthy or sick anti-CCP index is. Overall, however, the detection of anti-CCP AK also appears to be a parameter that is commonly determined. So what is new about the two cases described?
Author Response
We thank Reviewer 1 for his/her valuable comments. We hope that the following amendments, in particular in introduction, research design description and conclusions, have improved the quality of our manuscript and that it could now be considered for publication in Biomedicines.
Here follows the point-by-point response to the reviewer’s comments:
- After all, the authors write that the RM is a rare event. Are there reliable figures on how often this occurs? You yourself describe 26 cases from the literature in 20 years. That is very few.
We have now mentioned in our manuscript the findings of the recent literature review done by Trabelsi and coll. (already cited in our manuscript as ref. 3), reporting that less than 100 cases of RM are described in the literature [lines 41-42]. Regarding the 26 cases that we found in the literature, we reported that these were limited to those RM cases with measurement of RF and/or anti-CCP antibodies in CSF.
- As controls, wouldn't RA patients without proven RM be better?
We totally agree with the reviewer’s observation. Indeed, it came to our observation a woman with RA and a neurological manifestation different from RM (suspected neurodegenerative disorder) who underwent a lumbar puncture for diagnostic purposes: although serum anti-CCP antibodies value was elevated, her anti-CCP antibody index was negative, as reported in the manuscript. This is the only patients we had as control. We are actually collecting CSF and sera from patients affected by RA and other neurological diseases to obtain a cohort of controls, however this will take time.
- This also raises the question of how high the anti-CCP AK value is in the CSF of these patients. This would be important evidence for the significance of the anti-CCP titer. The same is true for neurological inflammatory diseases like multiple sclerosis. Here a comparison would be very interesting.
A polyspecific immune response known as MRZ reaction has been observed in a group of MS patients [lines 108-110]. Concerning anti CCP antibody titer in MS patients, we found only one study testing serum anti CCP titers with negative results (Alpayci M, Biomed Res Int. 2015).
- It would have to be worked out how high a healthy or sick anti-CCP index is.
In the manuscript we did not explain exhaustively the intrathecal synthesis process and what the Reiber’s formulas calculate. We have now tried to better elucidate the meaning of the IgG anti specific protein (here anti CCP) intrathecal production and the Antibody Index: the higher the amount of IgG anti CCP produced within the CNS, the higher the anti CCP Index value [lines 100-110].
- Overall, however, the detection of anti-CCP AK also appears to be a parameter that is commonly determined. So what is new about the two cases described?
As a matter of fact, while AI calculation for viral and some autoimmune diseases involving the CNS and for MRZ evaluation in MS is commonly used, anti CCP antibody index has been evaluated only in our study and in another one (Higashida-Konishi et al, 2020). A recent publication exhaustively explains the importance of this approach (Shamier MC, et al. The role of antibody indexes in clinical virology. Clin Microbiol Infect. 2021 Sep;27(9):1207-1211). In our opinion, anti CCP AI could be useful for diagnosis and monitoring of RM, since it is more reliable and accurate than value of anti CCP antibodies in CSF itself and less invasive than biopsy. This is now better discussed in our conclusions.
Reviewer 2 Report
General:
This is a very interesting and well written presentation. The limitations are those were only two cases and difficult to generalize but provide interesting findings for future studies. Clearly the application of index calculation offers a potential advantage over serum or CSF alone.
Specific:
Does the blood brain barrier become impaired in patients with RM.
L 23: were measured (grammar)
L 91: define QLim
L 134: What CSF inflammatory markers the are referring to in addition to the oligoclonal bands.
Table: The first patient had a positive oligoclonal band despite negative ant-CCP index and declining anti-CCP serum levels. Could the oligoclonal be an overread.
L 235: It is no clear if the authors calculated the anti-CCP index from literature data or merely reporting the isolated findings of serum and CSF data without the index. This need to be clarified. Additionally, it is not clear why this is considered part of the results section. Sounds more of a discussion on the published literature and thus should be in the discussion section.
Author Response
We thank Reviewer 2 for his/her valuable comments. We hope that the following amendments, in particular in research design and method description, have improved the quality of our manuscript and that it could now be considered for publication in Biomedicines.
Here follows the point-by-point response to the reviewer’s comments:
- Does the blood brain barrier become impaired in patients with RM?
While we did not detect BBB damage in our patients, this is reported in more or less 60% of cases that we analysed from the literature (data not shown).
- L 23: were measured (grammar)
Grammar mistake corrected.
- L 91: define QLim
We apologize for the imprecision, QLim has now been defined [lines 98].
- L 134: What CSF inflammatory markers they are referring to in addition to the oligoclonal bands?
Thanks for this observation. We generally consider the increase of CSF total proteins and cells as potential markers of inflammation. Some editing has been done to specify this point (see line 146).
- Table: The first patient had a positive oligoclonal band despite negative ant-CCP index and declining anti-CCP serum levels. Could the oligoclonal be an overread?
This is also a good point. We revised the images of the isoelectric focusing analytical session of the Case 1 and we actually confirm the presence of oligoclonal bands at T0 and T2.
- L 235: It is no clear if the authors calculated the anti-CCP index from literature data or merely reporting the isolated findings of serum and CSF data without the index. This need to be clarified. Additionally, it is not clear why this is considered part of the results section. Sounds more of a discussion on the published literature and thus should be in the discussion section.
As a matter of fact, we just reported previously published data we found in PubMed. From data available in these papers the calculation of the anti-CCP AI was not possible with the exception of Higashida-Konishi et al. that used exactly the same formulas. We move the whole paragraph (slightly edited) in discussion as suggested.
Round 2
Reviewer 1 Report
No further comments.